# VARIANCE REDUCTION FOR REINFORCEMENT LEARNING IN INPUT-DRIVEN ENVIRONMENTS

**Hongzi Mao, Shaileshh Bojja Venkatakrishnan, Malte Schwarzkopf, Mohammad Alizadeh**
MIT Computer Science and Artificial Intelligence Laboratory
{hongzi,bjjvnkt,malte,alizadeh}@csail.mit.edu

## ABSTRACT

We consider reinforcement learning in input-driven environments, where an exogenous, stochastic input process affects the dynamics of the system. Input processes arise in many applications, including queuing systems, robotics control with disturbances, and object tracking. Since the state dynamics and rewards depend on the input process, the state alone provides limited information for the expected future returns. Therefore, policy gradient methods with standard state-dependent baselines suffer high variance during training. We derive a bias-free, input-dependent baseline to reduce this variance, and analytically show its benefits over state-dependent baselines. We then propose a meta-learning approach to overcome the complexity of learning a baseline that depends on a long sequence of inputs. Our experimental results show that across environments from queuing systems, computer networks, and MuJoCo robotic locomotion, input-dependent baselines consistently improve training stability and result in better eventual policies.

## 1 INTRODUCTION

Deep reinforcement learning (RL) has emerged as a powerful approach for sequential decision-making problems, achieving impressive results in domains such as game playing (Mnih et al., 2015; Silver et al., 2017) and robotics (Levine et al., 2016; Schulman et al., 2015a; Lillicrap et al., 2015). This paper concerns RL in *input-driven environments*. Informally, input-driven environments have dynamics that are partially dictated by an exogenous, stochastic *input process*. Queuing systems (Kleinrock, 1976; Kelly, 2011) are an example; their dynamics are governed by not only the decisions made within the system (e.g., scheduling, load balancing) but also the arrival process that brings work (e.g., jobs, customers, packets) into the system. Input-driven environments also arise naturally in many other domains: network control and optimization (Winstein & Balakrishnan, 2013; Mao et al., 2017), robotics control with stochastic disturbances (Pinto et al., 2017), locomotion in environments with complex terrains and obstacles (Heess et al., 2017), vehicular traffic control (Belletti et al., 2018; Wu et al., 2017), tracking moving targets, and more (see Figure 1).

We focus on model-free policy gradient RL algorithms (Williams, 1992; Mnih et al., 2016; Schulman et al., 2015a), which have been widely adopted and benchmarked for a variety of RL tasks (Duan et al., 2016; Wu & Tian, 2017). A key challenge for these methods is the high variance in the gradient estimates, as such variance increases sample complexity and can impede effective learning (Schulman et al., 2015b; Mnih et al., 2016). A standard approach to reduce variance is to subtract a "baseline" from the total reward (or "return") to estimate the policy gradient (Weaver & Tao, 2001). The most common choice of a baseline is the *value function* — the expected return starting from the state.

Our main insight is that a state-dependent baseline — such as the value function — is a poor choice in input-driven environments, whose state dynamics and rewards are partially dictated by the input process. In such environments, comparing the return to the value function baseline may provide limited information about the quality of actions. The return obtained after taking a good action may be poor (lower than the baseline) if the input sequence following the action drives the system to unfavorable states; similarly, a bad action might end up with a high return with an advantageous input sequence. Intuitively, a good baseline for estimating the policy gradient should take the specific instance of the input process — the sequence of input values — into account. We call such a baseline an *input-dependent baseline*; it is a function of both the state and the entire future input sequence.

We formally define input-driven Markov decision processes, and we prove that an input-dependent baseline does not introduce bias in standard policy gradient algorithms such as Advantage Actor

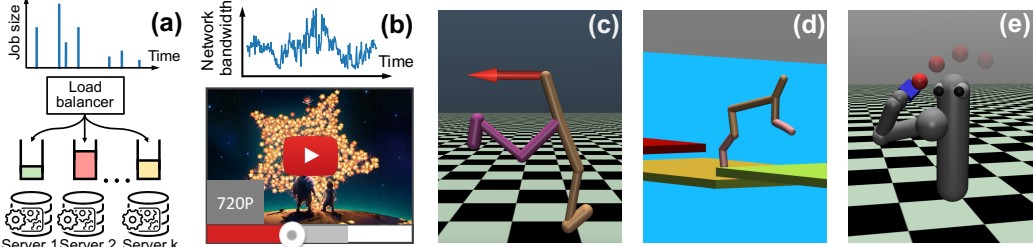

**Figure 1:** Input-driven environments: *(a)* load-balancing heterogeneous servers (Harchol-Balter & Vesilo, 2010) with stochastic job arrival as the input process; *(b)* adaptive bitrate video streaming (Mao et al., 2017) with stochastic network bandwidth as the input process; *(c)* Walker2d in wind with a stochastic force (wind) applied to the walker as the input process; *(d)* HalfCheetah on floating tiles with the stochastic process that controls the buoyancy of the tiles as the input process; *(e)* 7-DoF arm tracking moving target with the stochastic target position as the input process. Environments *(c)–(e)* use the MuJoCo physics simulator (Todorov et al., 2012).

Critic (A2C) (Mnih et al., 2016) and Trust Region Policy Optimization (TRPO) (Schulman et al., 2015a), provided that the input process is independent of the states and actions. We derive the optimal input-independent baseline and a simpler one to work with in practice; this takes the form of a *conditional value function* — the expected return given the state and the future input sequence.

Input-dependent baselines are harder to learn than their state-dependent counterparts; they are high-dimensional functions of the sequence of input values. To learn input-dependent baselines efficiently, we propose a simple approach based on meta-learning (Finn et al., 2017; Vilalta & Drissi, 2002). The idea is to learn a "meta baseline" that can be specialized to a baseline for a specific input instantiation using a small number of training episodes with that input. This approach applies to applications in which an input sequence can be repeated during training, e.g., applications that use simulations or experiments with previously-collected input traces for training (McGough et al., 2017).

We compare our input-dependent baseline to the standard value function baseline for the five tasks illustrated in Figure 1. These tasks are derived from queuing systems (load balancing heterogeneous servers (Harchol-Balter & Vesilo, 2010)), computer networks (bitrate adaptation for video streaming (Mao et al., 2017)), and variants of standard continuous control RL benchmarks in the MuJoCo physics simulator (Todorov et al., 2012). We adapted three widely-used MuJoCo benchmarks (Duan et al., 2016; Clavera et al., 2018a; Heess et al., 2017) to add a stochastic input element that makes these tasks significantly more challenging. For example, we replaced the static target in a 7-DoF robotic arm target-reaching task with a randomly-moving target that the robot aims to track over time. Our results show that input-dependent baselines consistently provide improved training stability and better eventual policies. Input-dependent baselines are applicable to a variety of policy gradient methods, including A2C, TRPO, PPO, robust adversarial RL methods such as RARL (Pinto et al., 2017), and meta-policy optimization such as MB-MPO (Clavera et al., 2018b). Video demonstrations of our experiments are available at `https://sites.google.com/view/input-dependent-baseline/`.

## 2 PRELIMINARIES

**Notation.** We consider a discrete-time Markov decision process (MDP), defined by $(\mathcal{S}, \mathcal{A}, \mathcal{P}, \rho_0, r, \gamma)$, where $\mathcal{S} \subseteq \mathbb{R}^n$ is a set of $n$-dimensional states, $\mathcal{A} \subseteq \mathbb{R}^m$ is a set of $m$-dimensional actions, $\mathcal{P} : \mathcal{S} \times \mathcal{A} \times \mathcal{S} \to [0, 1]$ is the state transition probability distribution, $\rho_0 : \mathcal{S} \to [0, 1]$ is the distribution over initial states, $r : \mathcal{S} \times \mathcal{A} \to \mathbb{R}$ is the reward function, and $\gamma \in (0, 1)$ is the discount factor. We denote a stochastic policy as $\pi : \mathcal{S} \times \mathcal{A} \to [0, 1]$, which aims to optimize the expected return $\eta(\pi) = \mathbb{E}_\tau \left[ \sum_{t=0}^{\infty} \gamma^t r(s_t, a_t) \right]$, where $\tau = (s_0, a_0, ...)$ is the trajectory following $s_0 \sim \rho_0$, $a_t \sim \pi(a_t|s_t)$, $s_{t+1} \sim \mathcal{P}(s_{t+1}|s_t, a_t)$. We use $V_\pi(s_t) = \mathbb{E}_{a_t, s_{t+1}, a_{t+1}, ...} \left[ \sum_{l=0}^{\infty} \gamma^l r(s_{t+l}, a_{r+l})|s_t \right]$ to define the value function, and $Q_\pi(s_t, a_t) = \mathbb{E}_{s_{t+1}, a_{t+1}, ...} \left[ \sum_{l=0}^{\infty} \gamma^l r(s_{t+l}, a_{r+l})|s_t, a_t \right]$ to define the state-action value function. For any sequence $(x_0, x_1, ...)$, we use $\boldsymbol{x}$ to denote the entire sequence and $x_{i:j}$ to denote $(x_i, x_{i+1}, ..., x_j)$.

**Policy gradient methods.** Policy gradient methods estimate the gradient of expected return with respect to the policy parameters (Sutton et al., 2000; Kakade, 2002; Gu et al., 2017). To train a policy

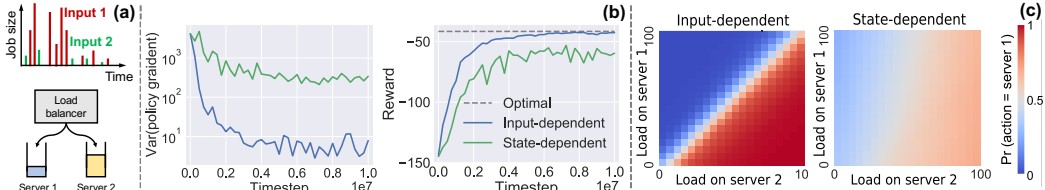

**Figure 2:** Load balancing over two servers. *(a)* Job sizes follow a Pareto distribution and jobs arrive as a Poisson process; the RL agent observes the queue lengths and picks a server for an incoming job. *(b)* The input-dependent baseline (blue) results in a $50\times$ lower policy gradient variance (left) and a 33% higher test reward (right) than the standard, state-dependent baseline (green). *(c)* The probability heatmap of picking server 1 shows that using the input-dependent baseline (left) yields a more precise policy than using the state-dependent baseline (right).

$\pi_\theta$ parameterized by $\theta$, the Policy Gradient Theorem (Sutton et al., 2000) states that

$$\nabla_\theta \eta(\pi_\theta) = \mathbb{E}_{\substack{s \sim \rho_\pi \\ a \sim \pi_\theta}} \left[ \nabla_\theta \log \pi_\theta(a|s) Q_{\pi_\theta}(s, a) \right], \tag{1}$$

where $\rho_\pi(s) = \sum_{t=0}^{\infty} \left[ \gamma^t \Pr(s_t = s) \right]$ denotes the discounted state visitation frequency. Practical algorithms often use the undiscounted state visitation frequency (i.e., $\gamma = 1$ in $\rho_\pi$), which can make the estimation slightly biased (Thomas, 2014).

Estimating the policy gradient using Monte Carlo estimation for the $Q$ function suffers from high variance (Mnih et al., 2016). To reduce variance, an appropriately chosen baseline $b(s_t)$ can be subtracted from the Q-estimate without introducing bias (Greensmith et al., 2004). The policy gradient estimation with a baseline in Equation (1) becomes $\mathbb{E}_{\rho_\pi, \pi_\theta} \left[ \nabla_\theta \log \pi_\theta(a|s) \left( Q_{\pi_\theta}(s, a) - b(s) \right) \right]$. While an optimal baseline exists (Greensmith et al., 2004; Wu et al., 2018), it is hard to estimate and often replaced by the value function $b(s_t) = V_\pi(s_t)$ (Sutton & Barto, 2017; Mnih et al., 2016).

## 3 MOTIVATING EXAMPLE

We use a simple load balancing example to illustrate the variance introduced by an exogenous input process. As shown in Figure 2a, jobs arrive over time and a load balancing agent sends them to one of two servers. The jobs arrive according to a Poisson process, and the job sizes follow a Pareto distribution. The two servers process jobs from their queues at identical rates. On each job arrival, the load balancer observes state $s_t = (q_1, q_2)$, denoting the queue length at the two servers. It then takes an action $a_t \in \{1, 2\}$, sending the job to one of the servers. The goal of the load balancer is to minimize the average job completion time. The reward corresponding to this goal is $r_t = -\tau \times j$, where $\tau$ is the time elapsed since the last action and $j$ is total number of enqueued jobs.

In this example, the optimal policy is to send the job to the server with the shortest queue (Daley, 1987). However, we find that a standard policy gradient algorithm, A2C (Mnih et al., 2016), trained using a value function baseline struggles to learn this policy. The reason is that the stochastic sequence of job arrivals creates huge variance in the reward signal, making it difficult to distinguish between good and bad actions. Consider, for example, an action at the state shown in Figure 2a. If the arrival sequence following this action consists of a burst of large jobs (e.g., input sequence 1 in Figure 2a), the queues will build up, and the return will be poor compared to the value function baseline (average return from the state). On the other hand, a light stream of jobs (e.g., input sequence 2 in Figure 2a) will lead to short queues and a better-than-average return. Importantly, this difference in return has little to do with the action; it is a consequence of the random job arrival process.

We train two A2C agents (Mnih et al., 2016), one with the standard value function baseline and the other with an input-dependent baseline tailored for each specific instantiation of the job arrival process (details of this baseline in §4). Since the the input-dependent baseline takes each input sequence into account explicitly, it reduces the variance of the policy gradient estimation much more effectively (Figure 2b, left). As a result, even in this simple example, only the policy learned with the input-dependent baseline comes close to the optimal (Figure 2b, right). Figure 2c visualizes the policies learned using the two baselines. The optimal policy (pick-shortest-queue) corresponds to a clear divide between the chosen servers at the diagonal.

In fact, the variance of the standard baseline can be arbitrarily worse than an input-dependent baseline: we refer the reader to Appendix A for an analytical example on a 1D grid world.

## 4 REDUCING VARIANCE FOR INPUT-DRIVEN MDPS

We now formally define input-driven MDPs and derive variance-reducing baselines for policy gradient methods in environments with input processes.

**Definition 1.** *An input-driven MDP is defined by* $(\mathcal{S}, \mathcal{A}, \mathcal{Z}, \mathcal{P}_s, \mathcal{P}_z, \rho_0^s, \rho_0^z, r, \gamma)$*, where* $\mathcal{Z} \subseteq \mathbb{R}^k$ *is a set of $k$-dimensional input values,* $\mathcal{P}_s(s_{t+1}|s_t, a_t, z_t)$ *is the transition kernel of the states,* $\mathcal{P}_z(z_{t+1}|z_{0:t})$ *is the transition kernel of the input process,* $\rho_0^z(z_0)$ *is the distribution of the initial input,* $r(s_t, a_t, z_t)$ *is the reward function, and* $\mathcal{S}$*,* $\mathcal{A}$*,* $\rho_0^s$*,* $\gamma$ *follow the standard definition in §2.*

An input-driven MDP adds an *input process*, $\boldsymbol{z} = (z_0, z_1, \cdots)$, to a standard MDP. In this setting, the next state $s_{t+1}$ depends on $(s_t, a_t, z_t)$. We seek to learn policies that maximize cumulative expected rewards. We focus on two cases, corresponding to the graphical models shown in Figure 3:

*Case 1: $z_t$ is a Markov process, and $\omega_t = (s_t, z_t)$ is observed at time $t$.* The action $a_t$ can hence depend on both $s_t$ and $z_t$.

*Case 2: $z_t$ is a general process (not necessarily Markov), and $\omega_t = s_t$ is observed at time $t$.* The action $a_t$ hence depends only on $s_t$.

In Appendix B, we prove that case 1 corresponds to a fully-observable MDP. This is evident from the graphical model in Figure 3a by considering $\omega_t = (s_t, z_t)$ to be the 'state' of the MDP at time $t$. Case 2, on the other hand, corresponds to a partially-observed MDP (POMDP) if we define the state to contain both $s_t$ and $z_{0:t}$, but leave $z_{0:t}$ *unobserved* at time $t$ (see Appendix B for details).

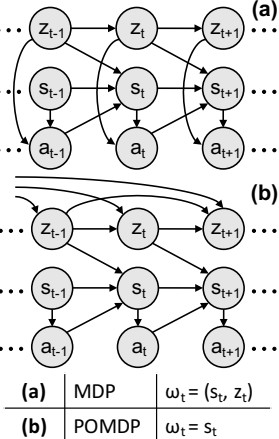

**Figure 3:** Graphical model of input-driven MDPs.

### 4.1 VARIANCE REDUCTION

In input-driven MDPs, the standard input-agnostic baseline is ineffective at reducing variance, as shown by our motivating example (§3). We propose to use an *input-dependent* baseline of the form $b(\omega_t, z_{t:\infty})$ — a function of both the observation at time $t$ and the input sequence from $t$ onwards. An input-dependent baseline uses information that is not available to the policy. Specifically, the input sequence $z_{t:\infty}$ cannot be used when taking an action at time $t$, because $z_{t+1:\infty}$ has not yet occurred at time $t$. However, in many applications, the input sequence is known at training time. In some cases, we know the entire input sequence upfront, e.g., when training in a simulator. In other situations, we can record the input sequence on the fly during training. Then, after a training episode, we can use the recorded values, including those that occurred after time $t$, to compute the baseline for each step $t$.

We now analyze input-dependent baselines. Our main result is that input-dependent baselines are bias-free. We also derive the optimal input-dependent baseline for variance reduction. All the results hold for both cases in Figure 3. We first state two useful lemmas required for our analysis. The first lemma shows that under the input-driven MDP definition, the input sequence $z_{t:\infty}$ is conditionally independent of the action $a_t$ given the observation $\omega_t$, while the second lemma states the policy gradient theorem for input-driven MDPs.

**Lemma 1.** $\Pr(z_{t:\infty}, a_t|\omega_t) = \Pr(z_{t:\infty}|\omega_t)\pi_\theta(a_t|\omega_t)$*, i.e., $z_{t:\infty} - \omega_t - a_t$ forms a Markov chain.*

*Proof.* See Appendix C.

**Lemma 2.** *For an input-driven MDP, the policy gradient theorem can be rewritten as*

$$\nabla_\theta \eta(\pi_\theta) = \mathbb{E}_{\substack{(\omega, \boldsymbol{z}) \sim \rho_\pi \\ a \sim \pi_\theta}} \left[ \nabla_\theta \log \pi_\theta(a|\omega) Q(\omega, a, \boldsymbol{z}) \right], \quad (2)$$

*where* $\rho_\pi(\omega, \boldsymbol{z}) = \sum_{t=0}^\infty \left[ \gamma^t \Pr(\omega_t = \omega, z_{t:\infty} = \boldsymbol{z}) \right]$ *denotes the discounted visitation frequency of the observation $\omega$ and input sequence $\boldsymbol{z}$, and* $Q(\omega, a, \boldsymbol{z}) = \mathbb{E} \left[ \sum_{l=0}^\infty \gamma^l r_{t+l} \mid \omega_t = \omega, a_t = a, z_{t:\infty} = \boldsymbol{z} \right]$.

*Proof.* See Appendix D.

Equation (2) generalizes the standard Policy Gradient Theorem in Equation (1). $\rho_\pi(\omega, \boldsymbol{z})$ can be thought of as a joint distribution over observations and input sequences. $Q(\omega, a, \boldsymbol{z})$ is a "state-action-input" value function, i.e., the expected return when taking action $a$ after observing $\omega$, with input sequence $\boldsymbol{z}$ from that step onwards. The key ingredient in the proof of Lemma 2 is the conditional independence of the input process $z_{t:\infty}$ and the action $a_t$ given the observation $\omega_t$ (Lemma 1).

**Theorem 1.** *An input-dependent baseline does not bias the policy gradient.*

*Proof.* Using Lemma 2, we need to show: $\mathbb{E}_{(\omega,\boldsymbol{z})\sim\rho_\pi,a\sim\pi_\theta}\left[\nabla_\theta\log\pi_\theta(a|\omega)b(\omega,\boldsymbol{z})\right]=0$. We have:

$$\mathbb{E}_{\substack{(\omega,\boldsymbol{z})\sim\rho_\pi\\a\sim\pi_\theta}}\left[\nabla_\theta\log\pi_\theta(a|\omega)b(\omega,\boldsymbol{z})\right]=\sum_\omega\sum_{\boldsymbol{z}}\sum_a\rho_\pi(\omega,\boldsymbol{z})\pi_\theta(a|\omega)\nabla_\theta\log\pi_\theta(a|\omega)b(\omega,\boldsymbol{z})$$

$$=\sum_\omega\sum_{\boldsymbol{z}}\rho_\pi(\omega,\boldsymbol{z})b(\omega,\boldsymbol{z})\sum_a\pi_\theta(a|\omega)\nabla_\theta\log\pi_\theta(a|\omega). \quad (3)$$

Since $\sum_a\pi_\theta(a|\omega)\nabla_\theta\log\pi_\theta(a|\omega)=\sum_a\nabla_\theta\pi_\theta(a|\omega)=\nabla_\theta\sum_a\pi_\theta(a|\omega)=0$, the theorem follows. □

Input-dependent baselines are also bias-free for policy optimization methods such as TRPO (Schulman et al., 2015a), as we show in Appendix F. Next, we derive the optimal input-dependent baseline for variance reduction. As the gradient estimates are vectors, we use the trace of the covariance matrix as the minimization objective (Greensmith et al., 2004).

**Theorem 2.** *The input-dependent baseline that minimizes variance in policy gradient is given by*

$$b^*(\omega,\boldsymbol{z})=\frac{\mathbb{E}_{a\sim\pi_\theta}\left[\nabla_\theta\log\pi_\theta(a|\omega)^T\nabla_\theta\log\pi_\theta(a|\omega)Q(\omega,a,\boldsymbol{z})\right]}{\mathbb{E}_{a\sim\pi_\theta}\left[\nabla_\theta\log\pi_\theta(a|\omega)^T\nabla_\theta\log\pi_\theta(a|\omega)\right]}. \quad (4)$$

*Proof.* See Appendix E.

Operationally, for observation $\omega_t$ at each step $t$, the input-dependent baseline takes the form $b(\omega_t,z_{t:\infty})$. In practice, we use a simpler alternative to Equation (4): $b(\omega_t,z_{t:\infty})=\mathbb{E}_{a_t\sim\pi_\theta}\left[Q(\omega_t,a_t,z_{t:\infty})\right]$. This can be thought of as a value function $V(\omega_t,z_{t:\infty})$ that provides the expected return given observation $\omega_t$ and input sequence $z_{t:\infty}$ from that step onwards. We discuss how to estimate input-dependent baselines efficiently in §5.

**Remark.** Input-dependent baselines are generally applicable to reducing variance for policy gradient methods in input-driven environments. In this paper, we apply input-dependent baselines to A2C (§6.2), TRPO (§6.1) and PPO (Appendix L). Our technique is complementary and orthogonal to adversarial RL (e.g., RARL (Pinto et al., 2017)) and meta-policy adaptation (e.g., MB-MPO (Clavera et al., 2018b)) for environments with external disturbances. Adversarial RL improves policy robustness by co-training an "adversary" to generate a worst-case disturbance process. Meta-policy optimization aims for fast policy adaptation to handle model discrepancy between training and testing. By contrast, input-dependent baselines improve policy optimization *itself* in the presence of stochastic input processes. Our work primarily focuses on learning a single policy in input-driven environments, without policy adaptation. However, input-dependent baselines can be used as a general method to improve the policy optimization step in adversarial RL and meta-policy adaptation methods. For example, in Appendix M, we empirically show that if an adversary generates high-variance noise, RARL with a standard state-based baseline cannot train good controllers, but the input-dependent baseline helps improve the policy's performance. Similarly, input-dependent baselines can improve meta-policy optimization in environments with stochastic disturbances, as we show in Appendix N.

## 5 LEARNING INPUT-DEPENDENT BASELINES EFFICIENTLY

Input-dependent baselines are functions of the sequence of input values. A natural approach to train such baselines is to use models that operate on sequences (e.g., LSTMs (Gers et al., 1999)). However, learning a sequential mapping in a high-dimensional space can be expensive (Bahdanau et al., 2014). We considered an LSTM approach, but ruled it out when initial experiments showed that it fails to provide significant policy improvement over the standard baseline in our environments (Appendix G).

Fortunately, we can learn the baseline much more efficiently in applications where we can repeat the same input sequence multiple times during training. Input-repeatability is feasible in many applications: it is straightforward when using simulators for training, and also feasible when training a real system with previously-collected input traces outside simulation. For example, training a robot in the presence of exogenous forces might apply a set of time-series traces of these forces repeatedly to the physical robot. We now present two approaches that exploit input-repeatability to learn input-dependent baselines efficiently.

**Multi-value-network approach.** A straightforward way to learn $b(\omega_t, z_{t:\infty})$ for different input instantiations $z$ is to train one value network to each particular instantiation of the input process. Specifically, in the training process, we first generate $N$ input sequences $\{z_1, z_2, \cdots, z_N\}$ and restrict training *only* to those $N$ sequences. To learn a separate baseline function for each input sequence, we use $N$ value networks with independent parameters $\theta_{V_1}, \theta_{V_2}, \cdots, \theta_{V_N}$, and single policy network with parameter $\theta$. During training, we randomly sample an input sequence $z_i$, execute a rollout based on $z_i$ with the current policy $\pi_\theta$, and use the (state, action, reward) data to train the value network parameter $\theta_{V_i}$ and the policy network parameter $\theta$ (details in Appendix I).

**Meta-learning approach.** The multi-value-network approach does not scale if the task requires training over a large number of input instantiations to generalize. The number of inputs needed is environment-specific, and can depend on a variety of factors, such as the time horizon of the problem, the distribution of the input process, the relative magnitude of the variance due to the input process compared to other sources of randomness (e.g., actions). Ideally, we would like an approach that enables learning across many different input sequences. We present a method based on *meta-learning* to train with an unbounded number of input sequences. The idea is to use *all* (potentially infinitely many) input sequences to learn a "meta value network" model. Then, for each specific input sequence, we first customize the meta value network using a few example rollouts with that input sequence. We then compute the actual baseline values for training the policy network parameters, using the customized value network for the specific input sequence. Our implementation uses Model-Agnostic Meta-Learning (MAML) (Finn et al., 2017).

---

**Algorithm 1** Training a meta input-dependent baseline for policy-based methods.

---

**Require:** $\alpha$, $\beta$: meta value network step size hyperparameters
1: Initialize policy network parameters $\theta$ and meta-value-network parameters $\theta_V$
2: **while** not done **do**
3:     Generate a new input sequence $z$
4:     Sample $k$ rollouts $\mathcal{T}_1, \mathcal{T}_2, ..., \mathcal{T}_k$ using policy $\pi_\theta$ and input sequence $z$
5:     Adapt $\theta_V$ with the first $k/2$ rollouts: $\theta_V^1 = \theta_V - \alpha \nabla_{\theta_V} \mathcal{L}_{\mathcal{T}_{1:k/2}} [V_{\theta_V}]$
6:     Estimate baseline value $V_{\theta_V^1}(\omega_t)$ for $s_t \sim \mathcal{T}_{k/2:k}$ using adapted $\theta_V^1$
7:     Adapt $\theta_V$ with the second $k/2$ rollouts: $\theta_V^2 = \theta_V - \alpha \nabla_{\theta_V} \mathcal{L}_{\mathcal{T}_{k/2:k}} [V_{\theta_V}]$
8:     Estimate baseline value $V_{\theta_V^2}(\omega_t)$ for $s_t \sim \mathcal{T}_{1:k/2}$ using adapted $\theta_V^2$
9:     Update policy with Equation (2) using the values from line (6) and (8) as baseline
10:     Update meta value network: $\theta_V \leftarrow \theta_V - \beta \nabla_{\theta_V} \mathcal{L}_{k/2:k} \left[ V_{\theta_V^1} \right] - \beta \nabla_{\theta_V} \mathcal{L}_{1:k/2} \left[ V_{\theta_V^2} \right]$
11: **end while**

---

The pseudocode in Algorithm 1 depicts the training algorithm. We follow the notation of MAML, denoting the loss in the value function $V_{\theta_V}(\cdot)$ on a rollout $\mathcal{T}$ as $\mathcal{L}_{\mathcal{T}}[V_{\theta_V}] = \sum_{\omega_t, r_t \sim \mathcal{T}} \| V_{\theta_V}(\omega_t) - \sum_{t'=t}^{T} \gamma^{t'-t} r_t \|^2$. We perform rollouts $k$ times with the same input sequence $z$ (lines 3 and 4); we use the first $k/2$ rollouts to customize the meta value network for this instantiation of $z$ (line 5), and then apply the customized value network on the states of the other $k/2$ rollouts to compute the baseline for those rollouts (line 6); similarly, we swap the two groups of rollouts and repeat the same process (lines 7 and 8). We use different rollouts to adapt the meta value network and compute the baseline to avoid introducing extra bias to the baseline. Finally, we use the baseline values computed for each rollout to update the policy network parameters (line 9), and we apply the MAML (Finn et al., 2017) gradient step to update the meta value network model (line 10).

## 6 EXPERIMENTS

Our experiments demonstrate that input-dependent baselines provide consistent performance gains across multiple continuous-action MuJoCo simulated robotic locomotions and discrete-action environments in queuing systems and network control. We conduct experiments for both policy gradient methods and policy optimization methods (see Appendix K for details). The videos for our experiments are available at https://sites.google.com/view/input-dependent-baseline/.

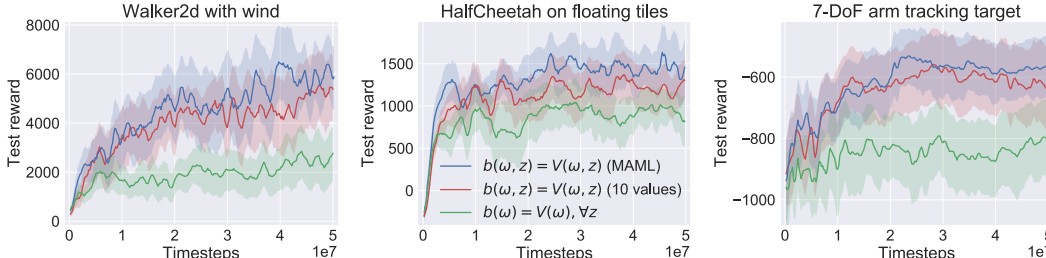

**Figure 4:** In continuous-action MuJoCo environments, TRPO (Schulman et al., 2015a) with input-dependent baselines achieve 25%–3× better testing reward than with a standard state-dependent baseline. Learning curves are on 100 testing episodes with unseen input sequences; shaded area spans one standard deviation.

## 6.1 SIMULATED ROBOTIC LOCOMOTION

We use the MuJoCo physics engine (Todorov et al., 2012) in OpenAI Gym (Brockman et al., 2016) to evaluate input-dependent baselines for robotic control tasks with external disturbance. We extend the standard Walker2d, HalfCheetah and 7-DoF robotic arm environments, adding a different external input to each (Figure 1).

**Walker2d with random wind (Figure 1c).** We train a 2D walker with varying wind, which randomly drags the walker backward or forward with different force at each step. The wind vector changes randomly, i.e., the wind forms a random input process. We add a force sensor to the state to enable the agent to quickly adapt. The goal is for the walker to walk forward while keeping balance.

**HalfCheetah on floating tiles with random buoyancy (Figure 1d).** A half-cheetah runs over a series of tiles floating on water (Clavera et al., 2018a). Each tile has different damping and friction properties, which moves the half-cheetah up and down and changes its dynamics. This random buoyancy is the external input process; the cheetah needs to learn running forward over varying tiles.

**7-DoF arm tracking moving target (Figure 1e).** We train a simulated robot arm to track a randomly moving target (a red ball). The robotic arm has seven degrees of freedom and the target is doing a random walk, which forms the external input process. The reward is the negative squared distance between the robot hand (blue square) and the target.

The Walker2d and 7-DoF arm environments correspond to the fully observable MDP case in Figure 3, i.e. the agent observes the input $z_t$ at time $t$. The HalfCheetah environment is a POMDP, as the agent does not observe the buoyancy of the tiles. In Appendix H, we show results for the POMDP version of the Walker2d environment.

**Results.** We build 10-value networks and a meta-baseline using MAML, both on top of the OpenAI's TRPO implementation (Dhariwal et al., 2017). Figure 4 shows the performance comparison among different baselines with 100 unseen testing input sequences at each training checkpoint. These learning curves show that TRPO with a state-dependent baseline performs worst in all environments. With the input-dependent baseline, by contrast, performance in unseen testing environments improves by up to 3×, as the agent learns a policy robust against disturbances. For example, it learns to lean into headwind and quickly place its leg forward to counter the headwind; it learns to apply different force on tiles with different buoyancy to avoid falling over; and it learns to co-adjust multiple joints to keep track of the moving object. The meta-baseline eventually outperforms 10-value networks as it effectively learns from a large number of input processes and hence generalizes better.

The input-dependent baseline technique applies generally on top of policy optimization methods. In Appendix L, we show a similar comparison with PPO (Schulman et al., 2017). Also, in Appendix M we show that adversarial RL (e.g., RARL (Pinto et al., 2017)) alone is not adequate to solve the high variance problem, and the input-dependent baseline helps improve the policy performance (Figure 9).

## 6.2 DISCRETE-ACTION ENVIRONMENTS

Our discrete-action environments arise from widely-studied problems in computer systems research: load balancing and bitrate adaptation.[1] As these problems often lack closed-form optimal solutions (Grandl et al., 2016; Yin et al., 2015), hand-tuned heuristics abound. Recent work suggests

---

[1]We considered Atari games often used as benchmark discrete-action RL environments (Mnih et al., 2015). However, Atari games lack an exogenous input process: a random seed perturbs the games' initial state, but it does not affect the environmental changes (e.g., in "Seaquest", the ships always come in a fixed pattern).

**Figure 5:** In environments with discrete action spaces, A2C (Mnih et al., 2016) with input-dependent baselines outperforms the best heuristic and achieves 25–33% better testing reward than vanilla A2C (Mnih et al., 2016). Learning curves are on 100 test episodes with unseen input sequences; shaded area spans one standard deviation.

that model-free reinforcement learning can achieve better performance than such human-engineered heuristics (Mao et al., 2016; Evans & Gao, 2016; Mao et al., 2017; Mirhoseini et al., 2017). We consider a load balancing environment (similar to the example in §3) and a bitrate adaptation environment in video streaming (Yin et al., 2015). The detailed setup of these environments is in Appendix J.

**Results.** We extend OpenAI's A2C implementation (Dhariwal et al., 2017) for our baselines. The learning curves in Figure 5 illustrate that directly applying A2C with a standard value network as the baseline results in unstable test reward and underperforms the traditional heuristic in both environments. Our input-dependent baselines reduce the variance and improve test reward by 25–33%, outperforming the heuristic. The meta-baseline performs the best in all environments.

## 7 RELATED WORK

Policy gradient methods compute unbiased gradient estimates, but can experience a large variance (Sutton & Barto, 2017; Weaver & Tao, 2001). Reducing variance for policy-based methods using a baseline has been shown to be effective (Williams, 1992; Sutton & Barto, 2017; Weaver & Tao, 2001; Greensmith et al., 2004; Mnih et al., 2016). Much of this work focuses on variance reduction in a general MDP setting, rather than variance reduction for MDPs with specific stochastic structures. Wu et al. (2018)'s techniques for MDPs with multi-variate independent actions are closest to our work. Their state-action-dependent baseline improves training efficiency and model performance on high-dimensional control tasks by explicitly factoring out, for each action, the effect due to other actions. By contrast, our work exploits the structure of state transitions instead of stochastic policy.

Recent work has also investigated the bias-variance tradeoff in policy gradient methods. Schulman et al. (2015b) replace the Monte Carlo return with a $\lambda$-weighted return estimation (similar to TD($\lambda$) with value function bootstrap (Tesauro, 1995)), improving performance in high-dimensional control tasks. Other recent approaches use more general control variates to construct variants of policy gradient algorithms. Tucker et al. (2018) compare the recent work, both analytically on a linear-quadratic-Gaussian task and empirically on complex robotic control tasks. Analysis of control variates for policy gradient methods is a well-studied topic, and extending such analyses (e.g., Greensmith et al. (2004)) to the input-driven MDP setting could be interesting future work.

In other contexts, prior work has proposed new RL training methodologies for environments with disturbances. Clavera et al. (2018b) adapts the policy to different pattern of disturbance by training the RL agent using meta-learning. RARL (Pinto et al., 2017) improves policy robustness by co-training an adversary to generate a worst-case noise process. Our work is orthogonal and complementary to these work, as we seek to improve policy optimization itself in the presence of inputs like disturbances.

## 8 CONCLUSION

We introduced input-driven Markov Decision Processes in which stochastic input processes influence state dynamics and rewards. In this setting, we demonstrated that an input-dependent baseline can significantly reduce variance for policy gradient methods, improving training stability and the quality of learned policies. Our work provides an important ingredient for using RL successfully in a variety of domains, including queuing networks and computer systems, where an input workload is a fundamental aspect of the system, as well as domains where the input process is more implicit, like robotics control with disturbances or random obstacles.

We showed that meta-learning provides an efficient way to learn input-dependent baselines for applications where input sequences can be repeated during training. Investigating efficient architectures for input-dependent baselines for cases where the input process cannot be repeated in training is an interesting direction for future work.

**Acknowledgements.** We thank Ignasi Clavera for sharing the HalfCheetah environment, Jonas Rothfuss for the comments on meta-policy optimization and the anonymous ICLR reviewers for their feedback. This work was funded in part by NSF grants CNS-1751009, CNS-1617702, a Google Faculty Research Award, an AWS Machine Learning Research Award, a Cisco Research Center Award, an Alfred P. Sloan Research Fellowship and the sponsors of MIT Data Systems and AI Lab.

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

# A    ILLUSTRATION OF VARIANCE REDUCTION IN 1D GRID WORLD

Consider a walker in a 1D grid world, where the state $s_t \in \mathbb{Z}$ at time $t$ denotes the position of the walker, and action $a_t \in \{-1, +1\}$ denotes the intent to either move forward or backward. Additionally let $z_t \in \{-1, +1\}$ be a uniform i.i.d. "exogenous input" that perturbs the position of the walker. For an action $a_t$ and input $z_t$, the state of the walker in the next step is given by $s_{t+1} = s_t + a_t + z_t$. The objective of the game is to move the walker forward; hence, the reward is $r_t = a_t + z_t$ at each time step. $\gamma \in [0, 1]$ is a discount factor.

While the optimal policy for this game is clear ($a_t = +1$ for all $t$), consider learning such a policy using policy gradient. For simplicity, let the policy be parametrized as $\pi_\theta(a_t = +1|s_t) = e^\theta/(1+e^\theta)$, with $\theta$ initialized to 0 at the start of training. In the following, we evaluate the variance of the policy gradient estimate at the start of training under (i) the standard value function baseline, and (ii) a baseline that is the expected cumulative reward conditioned on all future $z_t$ inputs.

**Variance under standard baseline.** The value function in this case is identically 0 at all states. This is because $\mathbb{E}[\sum_{t=0}^{\infty} \gamma^t r_t] = \mathbb{E}[\sum_{t=0}^{\infty} \gamma^t(a_t + z_t)] = 0$ since both actions $a_t$ and inputs $z_t$ are i.i.d. with mean 0. Also note that $\nabla_\theta \log \pi_\theta(a_t = +1) = 1/2$ and $\nabla_\theta \log \pi_\theta(a_t = -1) = -1/2$; hence $\nabla_\theta \log \pi_\theta(a_t) = a_t/2$. Therefore the variance of the policy gradient estimate can be written as

$$V_1 = \text{Var}\left[\sum_{t=0}^{\infty} \frac{a_t}{2} \sum_{t'=t}^{\infty} \gamma^{t'} r_{t'}\right] = \text{Var}\left[\sum_{t=0}^{\infty} \frac{a_t}{2} \sum_{t'=t}^{\infty} \gamma^{t'}(a_{t'} + z_{t'})\right]. \tag{5}$$

**Variance under input-dependent baseline.** Now, consider an alternative "input-dependent" baseline $V(s_t|\mathbf{z})$ defined as $\mathbb{E}[\sum_{t=0}^{\infty} \gamma^t r_t|\mathbf{z}]$. Intuitively this baseline captures the average reward incurred when experiencing a particular fixed $\mathbf{z}$ sequence. We refer the reader to §4 for a formal discussion and analysis of input-dependent baselines. Evaluating the baseline we get $V(s_t|\mathbf{z}) = \mathbb{E}[\sum_{t=0}^{\infty} \gamma^t r_t|\mathbf{z}] = \sum_{t=0}^{\infty} \gamma^t z_t$. Therefore the variance of the policy gradient estimate in this case is

$$V_2 = \text{Var}\left[\sum_{t=0}^{\infty} \frac{a_t}{2}\left(\sum_{t'=t}^{\infty} \gamma^{t'} r_{t'} - \sum_{t'=t}^{\infty} \gamma^{t'} z_{t'}\right)\right] = \text{Var}\left[\sum_{t=0}^{\infty} \frac{a_t}{2}\left(\sum_{t'=t}^{\infty} \gamma^{t'} a_{t'}\right)\right]. \tag{6}$$

**Reduction in variance.** To analyze the variance reduction between the two cases (Equations (5) and (6)), we note that

$$V_1 = V_2 + \text{Var}\left[\sum_{t=0}^{\infty} \frac{a_t}{2}\left(\sum_{t'=t}^{\infty} \gamma^{t'} z_{t'}\right)\right] + 2\text{Cov}\left(\sum_{t=0}^{\infty} \frac{a_t}{2}\left(\sum_{t'=t}^{\infty} \gamma^{t'} a_{t'}\right), \sum_{t=0}^{\infty} \frac{a_t}{2}\left(\sum_{t'=t}^{\infty} \gamma^{t'} z_{t'}\right)\right) \tag{7}$$

$$= V_2 + \text{Var}\left[\sum_{t=0}^{\infty} \frac{a_t}{2}\left(\sum_{t'=t}^{\infty} \gamma^{t'} z_{t'}\right)\right]. \tag{8}$$

This follows because

$$\mathbb{E}\left[\sum_{t=0}^{\infty} \frac{a_t}{2}\left(\sum_{t'=t}^{\infty} \gamma^{t'} z_{t'}\right)\right] = \sum_{t=0}^{\infty} \sum_{t'=t}^{\infty} \frac{\gamma^{t'}}{2} \mathbb{E}[a_t z_{t'}] = 0, \quad \text{and}$$

$$\mathbb{E}\left[\left(\sum_{t=0}^{\infty} \frac{a_t}{2}\left(\sum_{t'=t}^{\infty} \gamma^{t'} a_{t'}\right)\right)\left(\sum_{t=0}^{\infty} \frac{a_t}{2}\left(\sum_{t'=t}^{\infty} \gamma^{t'} z_{t'}\right)\right)\right] =$$

$$\sum_{t_1=0}^{\infty} \sum_{t_1'=t_1}^{\infty} \sum_{t_2=0}^{\infty} \sum_{t_2'=t_2}^{\infty} \mathbb{E}\left[\frac{a_{t_1} a_{t_1'} a_{t_2} z_{t_2'}}{4} \gamma^{t_1'+t_2'}\right] = 0.$$

Therefore the covariance term in Equation (7) is 0. Hence the variance reduction from Equation (8) can be written as

$$V_1 - V_2 = \text{Var}\left[\sum_{t=0}^{\infty} \frac{a_t}{2}\left(\sum_{t'=t}^{\infty} \gamma^{t'} z_{t'}\right)\right] = \sum_{t_1=0}^{\infty} \sum_{t_1'=t_1}^{\infty} \sum_{t_2=0}^{\infty} \sum_{t_2'=t_2}^{\infty} \mathbb{E}\left[\frac{a_{t_1} a_{t_2} z_{t_1'} z_{t_2'}}{4} \gamma^{t_1'+t_2'}\right]$$

$$= \sum_{t_1=0}^{\infty} \sum_{t_1'=t_1}^{\infty} \mathbb{E}\left[\frac{a_{t_1}^2 z_{t_1'}^2}{4} \gamma^{2t_1'}\right] = \frac{\text{Var}(a_0)\text{Var}(z_0)}{4(1-\gamma^2)^2}.$$

Thus the input-dependent baseline reduces variance of the policy gradient estimate by an amount proportional to the variance of the external input. In this toy example, we have chosen $z_t$ to be binary-valued, but more generally the variance of $z_t$ could be arbitrarily large and might be a dominating factor of the overall variance in the policy gradient estimation.

## B    MARKOV PROPERTIES OF INPUT-DRIVEN DECISION PROCESSES

**Proposition 1.** *An input-driven decision process satisfying the conditions of case 1 in Figure 3 is a fully observable MDP, with state $\tilde{s}_t := (s_t, z_t)$, and action $\tilde{a}_t := a_t$.*

*Proof.*

$$
\begin{aligned}
\Pr(\tilde{s}_{t+1}|\tilde{s}_{0:t}, \tilde{a}_{0:t}) &= \Pr(s_{t+1}, z_{t+1}|s_{0:t}, z_{0:t}, a_{0:t}) \\
&= \Pr(s_{t+1}, z_{t+1}|s_t, z_t, a_t) \quad \text{(by definition of case 1 in Figure 3a)} \\
&= \Pr(\tilde{s}_{t+1}|\tilde{s}_t, \tilde{a}_t).
\end{aligned}
$$

$\square$

**Proposition 2.** *An input-driven decision process satisfying the conditions of case 2 in Figure 3, with state $\tilde{s}_t := (s_t, z_{0:t})$ and action $\tilde{a}_t := a_t$ is a fully observable MDP. If only $\omega_t = s_t$ is observed at time t, it is a partially observable MDP (POMDP).*

*Proof.*

$$
\begin{aligned}
\Pr(\tilde{s}_{t+1}|\tilde{s}_{0:t}, \tilde{a}_{0:t}) &= \Pr(s_{t+1}, z_{0:t+1}|s_{0:t}, z_{0:t}, a_{0:t}) \\
&= \Pr(s_{t+1}|s_{0:t}, z_{0:t+1}, a_{0:t}) \Pr(z_{0:t+1}|s_{0:t}, z_{0:t}, a_{0:t}) \\
&= \Pr(s_{t+1}|s_t, z_{0:t+1}, a_t) \Pr(z_{0:t+1}|s_t, z_{0:t}, a_t) \text{ (by definition of case 2 in Figure 3b)} \\
&= \Pr(s_{t+1}, z_{0:t+1}|s_t, z_{0:t}, a_t) \\
&= \Pr(\tilde{s}_{t+1}|\tilde{s}_t, \tilde{a}_t).
\end{aligned}
$$

Therefore, $(\tilde{s}_t, \tilde{a}_t)$ is a fully observable MDP. If only $\omega_t = s_t$ is observed, the decision process is a POMDP, since the $z_{0:t}$ component of the state is not observed. $\square$

## C    PROOF OF LEMMA 1

*Proof.* From the definition of an input-driven MDP (Definition 1), we have

$$
\begin{aligned}
\Pr(z_{0:\infty}, \omega_t, a_t) &= \Pr(z_{0:t}, \omega_t, a_t) \Pr(z_{t+1:\infty}|z_{0:t}, \omega_t, a_t) \\
&= \Pr(z_{0:t}, \omega_t) \Pr(a_t|z_{0:t}, \omega_t) \Pr(z_{t+1:\infty}|z_{0:t}) \\
&= \Pr(z_{0:t}, \omega_t) \pi_\theta(a_t|\omega_t) \Pr(z_{t+1:\infty}|z_{0:t}) \\
&= \Pr(z_{0:\infty}, \omega_t) \pi_\theta(a_t|\omega_t). \quad\quad (9)
\end{aligned}
$$

Notice that $\Pr(a_t|z_{0:t}, \omega_t) = \pi_\theta(a_t|\omega_t)$ in both the MDP and POMDP cases in Figure 3. By marginalizing over $z_{0:t-1}$ on both sides, we obtain the result:

$$
\Pr(z_{t:\infty}, \omega_t, a_t) = \Pr(z_{t:\infty}, \omega_t) \pi_\theta(a_t|\omega_t). \quad\quad (10)
$$

$\square$

## D    PROOF OF LEMMA 2

*Proof.* Expanding the Policy Gradient Theorem (Sutton & Barto, 2017), we have

$$
\begin{aligned}
\nabla_\theta \eta(\pi_\theta) =& \mathbb{E}\left[\sum_{t=0}^{\infty} \nabla_\theta \log \pi_\theta(a_t|\omega_t) \sum_{t' \geq t} \gamma^{t'} r_{t'}\right] \\
=& \sum_{t=0}^{\infty} \mathbb{E}\left[\nabla_\theta \log \pi_\theta(a_t|\omega_t) \sum_{t' \geq t} \gamma^{t'} r_{t'}\right] \\
=& \sum_{t=0}^{\infty} \left[\sum_{\omega,a,\boldsymbol{z}} \Pr(\omega_t = \omega, a_t = a, z_{t:\infty} = \boldsymbol{z}) \nabla_\theta \log \pi_\theta(a|\omega) \mathbb{E}\left[\sum_{t' \geq t} \gamma^{t'} r_{t'}|\omega_t = \omega, a_t = a, z_{t:\infty} = \boldsymbol{z}\right]\right] \\
=& \sum_{t=0}^{\infty} \left[\sum_{\omega,a,\boldsymbol{z}} \Pr(\omega_t = \omega, z_{t:\infty} = \boldsymbol{z}) \pi_\theta(a|\omega) \nabla_\theta \log \pi_\theta(a|\omega) \mathbb{E}\left[\sum_{t' \geq t} \gamma^{t'} r_{t'}|\omega_t = \omega, a_t = a, z_{t:\infty} = \boldsymbol{z}\right]\right],
\end{aligned}
\tag{11}
$$

where the last step uses Lemma 1. Using the definition of $Q(\omega, a, \boldsymbol{z})$, we obtain:

$$
\begin{aligned}
\nabla_\theta \eta(\pi_\theta) =& \sum_{t=0}^{\infty}\left[\sum_{\omega,a,\boldsymbol{z}} \Pr(\omega_t = \omega, z_{t:\infty} = \boldsymbol{z}) \pi_\theta(a|\omega) \nabla_\theta \log \pi_\theta(a|\omega) \gamma^t Q(\omega, a, \boldsymbol{z})\right] \\
=& \sum_{\omega,a,\boldsymbol{z}}\left[\pi_\theta(a|\omega) \nabla_\theta \log \pi_\theta(a|\omega) Q(\omega, a, \boldsymbol{z}) \left[\sum_{t=0}^{\infty} \gamma^t \Pr(\omega_t = \omega, z_{t:\infty} = \boldsymbol{z})\right]\right] \\
=& \sum_{\omega,a,\boldsymbol{z}} \pi_\theta(a|\omega) \nabla_\theta \log \pi_\theta(a|\omega) Q(\omega, a, \boldsymbol{z}) \rho_\pi(\omega, \boldsymbol{z}) \\
=& \mathbb{E}_{\substack{(\omega,\boldsymbol{z}) \sim \rho_\pi \\ a \sim \pi_\theta}}\left[\nabla_\theta \log \pi_\theta(a|\omega) Q(\omega, a, \boldsymbol{z})\right].
\end{aligned}
\tag{12}
$$

$\square$

## E    PROOF OF THEOREM 2

*Proof.* Let $G(\omega, a)$ denote $\nabla_\theta \log \pi_\theta(a|\omega)^T \nabla_\theta \log \pi_\theta(a|\omega)$. For any input-dependent baseline $b(\omega, \boldsymbol{z})$, the variance of the policy gradient estimate is given by

$$
\mathbb{E}_{\substack{(\omega,\boldsymbol{z}) \sim \rho_\pi \\ a \sim \pi_\theta}}\left[\left\|\nabla_\theta \log \pi_\theta(a|\omega)\left[Q(\omega, a, \boldsymbol{z}) - b(\omega, \boldsymbol{z})\right] - \mathbb{E}_{\rho_\pi,\pi_\theta}\left[\nabla_\theta \log \pi_\theta(a|\omega)\left[Q(\omega, a, \boldsymbol{z}) - b(\omega, \boldsymbol{z})\right]\right]\right\|_2^2\right]
$$

$$
= \mathbb{E}_{\rho_\pi,\pi_\theta}\left[G(\omega, a)\left[Q(\omega, a, \boldsymbol{z}) - b(\omega, \boldsymbol{z})\right]^2\right] - \left\|\mathbb{E}_{\rho_\pi,\pi_\theta}\left[\nabla_\theta \log \pi_\theta(a|\omega)\left[Q(\omega, a, \boldsymbol{z}) - b(\omega, \boldsymbol{z})\right]\right]\right\|_2^2
$$

$$
= \mathbb{E}_{\rho_\pi,\pi_\theta}\left[G(\omega, a)\left[Q(\omega, a, \boldsymbol{z}) - b(\omega, \boldsymbol{z})\right]^2\right] - \left\|\mathbb{E}_{\rho_\pi,\pi_\theta}\left[\nabla_\theta \log \pi_\theta(a|\omega) Q(\omega, a, \boldsymbol{z})\right]\right\|_2^2 \quad \text{(due to Theorem 1)}
$$

$$
= \mathbb{E}_{\rho_\pi,\pi_\theta}\left[G(\omega, a) Q(\omega, a, \boldsymbol{z})^2\right] - \left\|\mathbb{E}_{\rho_\pi,\pi_\theta}\left[\nabla_\theta \log \pi_\theta(a|\omega) Q(\omega, a, \boldsymbol{z})\right]\right\|_2^2
$$

$$
+ \mathbb{E}_{\rho_\pi}\left[\mathbb{E}_{a \sim \pi_\theta}\left[G(\omega, a) \mid \boldsymbol{z}, \omega\right] b(\omega, \boldsymbol{z})^2 - 2\mathbb{E}_{a \sim \pi_\theta}\left[G(\omega, a) Q(\omega, a, \boldsymbol{z}) \mid \omega, \boldsymbol{z}\right] b(\omega, \boldsymbol{z})\right].
$$

Notice that the baseline is only involved in the last term in a quadratic form, where the second order term is positive. To minimize the variance, we set baseline to the minimizer of the quadratic equation, i.e., $2\mathbb{E}_{a \sim \pi_\theta}\left[G(\omega, a) \mid \omega, \boldsymbol{z}\right] b(\omega, \boldsymbol{z}) - 2\mathbb{E}_{a \sim \pi_\theta}\left[G(\omega, a) Q(\omega, a, \boldsymbol{z}) \mid \omega, \boldsymbol{z}\right] = 0$ and hence the result follows. $\square$

## F    INPUT-DEPENDENT BASELINE FOR TRPO

We show that the input-dependent baselines are bias-free for Trust Region Policy Optimization (TRPO) (Schulman et al., 2015a).

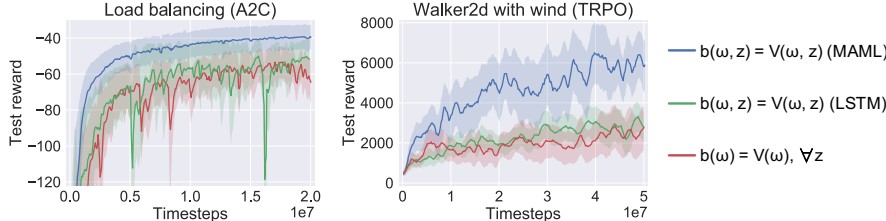

**Figure 6:** An LSTM-based input-dependent baseline (green) does not provide significant performance gain over standard state-dependent baseline (red) for load balancing and Walker2d with wind environments.

**Preliminaries.** Stochastic gradient descent using Equation (1) does not guarantee consistent policy improvement in complex control problems. TRPO is an alternative approach that offers monotonic policy improvements, and derives a practical algorithm with better sample efficiency and performance. TRPO maximizes a surrogate objective, subject to a KL divergence constraint:

$$\underset{\theta}{\text{maximize}} \quad \mathbb{E}_{\substack{s \sim \rho_{\pi_{\text{old}}} \\ a \sim \pi_{\text{old}}}} \left[ \frac{\pi_\theta(a|s)}{\pi_{\text{old}}(a|s)} Q_{\pi_{\text{old}}}(s, a) \right] \tag{13}$$

$$\text{subject to} \quad \mathbb{E}_{s \sim \rho_{\pi_{\text{old}}}} \left[ D_{\text{KL}} \left( \pi_{\text{old}}(\cdot|s) || \pi_\theta(\cdot|s) \right) \right] \le \delta, \tag{14}$$

in which $\delta$ serves as a step size for policy update. Using a baseline in the TRPO objective, i.e. replacing $Q_{\pi_{\text{old}}}(s, a)$ with $Q_{\pi_{\text{old}}}(s, a) - b(s)$, empirically improves policy performance (Schulman et al., 2015b).

Similar to Theorem 2, we generalize TRPO to input-driven environments, with $\rho_\pi(\omega, z) = \sum_{t=0}^{\infty} \left[ \gamma^t \Pr(\omega_t = \omega, z_{t:\infty} = z) \right]$ denoting the discounted visitation frequency of the observation $\omega$ and input sequence $z$, and $Q(\omega, a, z) = \mathbb{E} \left[ \sum_{l=0}^{\infty} \gamma^l r_{t+l} \mid \omega_t = \omega, a_t = a, z_{t:\infty} = z \right]$. The TRPO objective becomes $\mathbb{E}_{(\omega, z) \sim \rho_{\text{old}}, a \sim \pi_{\text{old}}} \left[ Q_{\pi_{\text{old}}}(\omega, a, z) \pi_\theta(a|\omega)/\pi_{\text{old}}(a|\omega) \right]$, and the constraint is $\mathbb{E}_{(\omega, z) \sim \rho_{\pi_{\text{old}}}} \left[ D_{\text{KL}} \left( \pi_{\text{old}}(\cdot|s) || \pi_\theta(\cdot|s) \right) \right] \le \delta$.

**Theorem 3.** *An input-dependent baseline does not change the optimal solution of the optimization problem in TRPO, that is $argmax_\theta \mathbb{E}_{(\omega, z) \sim \rho_{old}, a \sim \pi_{old}} \left[ \frac{\pi_\theta(a|\omega)}{\pi_{old}(a|\omega)} Q_{\pi_{old}}(\omega, a, z) \right] = argmax_\theta \mathbb{E}_{(\omega, z) \sim \rho_{old}, a \sim \pi_{old}} \left[ \frac{\pi_\theta(a|\omega)}{\pi_{old}(a|\omega)} \left( Q_{\pi_{old}}(\omega, a, z) - b(\omega, z) \right) \right]$.*

*Proof.*

$$\mathbb{E}_{(\omega, z) \sim \rho_{\text{old}}, a \sim \pi_{\text{old}}} \left[ \frac{\pi_\theta(a|\omega)}{\pi_{\text{old}}(a|\omega)} b(\omega, z) \right] = \sum_\omega \sum_z \rho_{\text{old}}(\omega, z) \sum_a \pi_{\text{old}}(a|\omega) \left[ \frac{\pi_\theta(a|\omega)}{\pi_{\text{old}}(a|\omega)} b(\omega, z) \right]$$

$$= \sum_\omega \sum_z \rho_{\text{old}}(\omega, z) \sum_a \pi_\theta(a|\omega) b(\omega, z)$$

$$= \sum_\omega \sum_z \rho_{\text{old}}(\omega, z) b(\omega, z),$$

which is independent of $\theta$. Therefore, $b(\omega, z)$ does not change the optimal solution to the optimization problem. $\square$

## G   INPUT-DEPENDENT BASELINE WITH LSTM

The input-dependent baseline is a function of both the state and the entire future input sequence. A natural approach to approximate such baselines is to use neural models that operate on sequences (e.g., LSTMs (Gers et al., 1999)). However, learning a sequential mapping in a high-dimensional space can be expensive (Bahdanau et al., 2014). For example, consider the LSTM input-dependent baseline with A2C on the load balancing environment (Figure 1a; §6.2) and TRPO on the Walker2d with wind environment (Figure 1c; §6.1). As shown in Figure 6, the LSTM input-dependent baseline (green) does not significantly improve the policy performance over a standard state-only baseline (red) in these environments. By contrast, the MAML based input-dependent baseline (§5) reduces the variance in policy gradient estimation much more effectively and achieves a consistently better policy performance.

## H    ADDITIONAL POMDP EXPERIMENT

Recall that the HalfCheetah on floating tiles environment (Figure 1d) is a POMDP, since the agent does not observe the buoyancy of the tiles. Figure 4 (middle) shows that the input-dependent baseline significantly improves the TRPO performance for this POMDP environment. We also created a POMDP version of the Walker2d with wind environment (Figure 1c), where the force of the wind is removed from the observation provided to the agent. The results of repeating the same training are shown in Figure 7. We make two key observations: (1) Compared with the MDP case in Figure 4 (left), the performance drops slightly overall. This is expected because the agent can react more agilely if it directly observes the current wind in the MDP case. (2) Input-dependent baselines reduce variance and improve the policy performance, with the MAML-based approach achieving the best performance, similar to the MDP case.

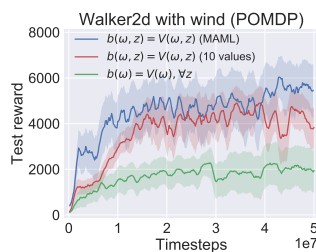

**Figure 7:** Input-dependent baseline improves TRPO performance in the POMDP version of the Walker2d with wind environment.

## I    PSEUDOCODE FOR TRAINING MULTI-VALUE BASELINES

In §5, we explained the idea of efficiently computing input-dependent baselines (§4.1) using multiple value networks on a fixed set of input sequences. Algorithm 2 depicts the details of this approach.

---

**Algorithm 2** Training multi-value baselines for policy-based methods.

---

**Require:**  pregenerated input seuqnces $\{z_1, z_2, \cdots, z_N\}$, step sizes $\alpha, \beta$
1: Initialize value network parameters $\theta_{V_1}, \theta_{V_1}, \cdots, \theta_{V_N}$ and policy parameters $\theta$
2: **while** not done **do**
3:     Sample a input sequence $z_i$
4:     Sample $k$ rollouts $\mathcal{T}_1, \mathcal{T}_2, ..., \mathcal{T}_k$ using policy $\pi_\theta$ and input sequence $z_i$
5:     Update policy with Equation (2) using baseline estimated with $\theta_{V_i}$
6:     Update $i$-th value network parameters: $\theta_{V_i} \leftarrow \theta_{V_i} - \beta \nabla_{\theta_{V_i}} \mathcal{L}_{1:k} \left[ V_{\theta_{V_i}} \right]$
7: **end while**

---

## J    SETUP FOR DISCRETE-ACTION ENVIRONMENTS

**Load balancing across servers (Figure 1a).** In this environment, an RL agent balances jobs over $k$ servers to minimize the average job completion time. Similar to §3, the job sizes follow a Pareto distribution (scale $x_m = 100$, shape $\alpha = 1.5$), and jobs arrive in a Poisson process ($\lambda = 55$). We run over 10 simulated servers with different processing rates, ranging linearly from 0.15 to 1.05. In this setting, the load of the system is at 90% (i.e., on average, 90% of the queues are non-empty). In each episode, we generate 500 jobs as the exogenous input process. The problem of minimizing average job completion time on servers with heterogeneous processing rates does not have a closed-form solution (Harchol-Balter & Vesilo, 2010); the most widely-used heuristic is to join the shortest queue (Daley, 1987). However, understanding the workload pattern can give a better policy; for example, we can reserve some servers for small jobs. In this environment, the observed state is a vector of $(j, q_1, q_2, ..., q_k)$, where $j$ is the size of the incoming job, $q_i$ is the amount of work currently in each queue. The action $a \in \{1, 2, ..., k\}$ schedules the incoming job to a specific queue. The reward is the number of active jobs times the negated time elapsed since the last action.

**Bitrate adaptation for video streaming (Figure 1b).** Streaming video over variable-bandwidth connections requires the client to adapt the video bitrates to optimize the user experience. This is challenging since the available network bandwidth (the exogenous input process) is hard to predict accurately. We simulate real-world video streaming using public cellular network data (Riiser et al., 2013) and video with seven bitrate levels and 500 chunks (DASH Industry Form, 2016). The reward is a weighted combination of video resolution, time paused for rebuffering, and the number of bitrate changes (Mao et al., 2017). The observed state contains bandwidth history, current video buffer size, and current bitrate. The action is the next video chunk's bitrate. State-of-the-art heuristics for this

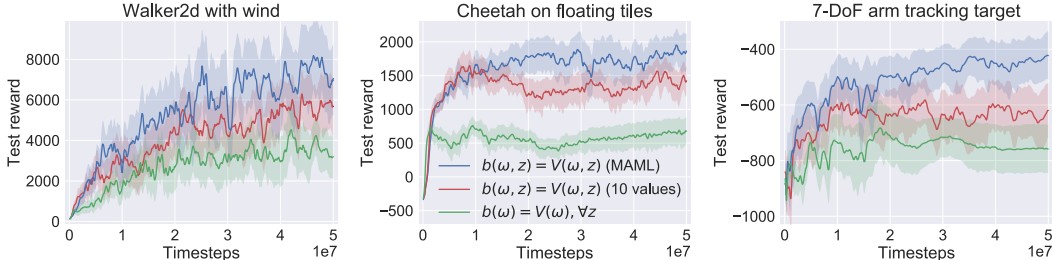

**Figure 8:** In continuous-action MuJoCo environments (§6.1), PPO (Schulman et al., 2017) with input-dependent baselines achieves 42%–3.5× better testing reward than PPO with a standard state-dependent baseline. Learning curves are on 100 testing episodes with unseen input sequences; shaded area spans one standard deviation.

problem conservatively estimate the network bandwidth and use model predictive control to choose the optimal bitrate over the near-term horizon (Yin et al., 2015).

## K   EXPERIMENT DETAILS

In our discrete-action environments (§6.2), we build 10-value networks and a meta-baseline using MAML (Finn et al., 2017), both on top of the OpenAI A2C implementation (Dhariwal et al., 2017). We use $\gamma = 0.995$ for both environments. The actor and the critic networks have 2 hidden layers, with 64 and 32 hidden neurons on each. The activation function is ReLU (Nair & Hinton, 2010) and the optimizer is Adam (Chilimbi et al., 2014). We train the policy with 16 (synchronous) parallel agents. The learning rate is $1^{-3}$. The entropy factor (Mnih et al., 2016) is decayed linearly from 1 to 0.001 over 10,000 training iterations. For the meta-baseline, the meta learning rate is $1^{-3}$ and the model specification has five step updates, each with learning rate $1^{-4}$. The model specification step in MAML is performed with vanilla stochastic gradient descent.

We introduce disturbance into our continuous-action robot control environments (§6.1). For the walker with wind (Figure 1c), we randomly sample a wind force in $[-1, 1]$ initially and add a Gaussian noise sampled from $\mathcal{N}(0, 1)$ at each step. The wind is bounded between $[-10, 10]$. The episode terminates when the walker falls. For the half-cheetah with floating tiles, we extend the number of piers from 10 in the original environment (Clavera et al., 2018a) to 50, so that the agent remains on the pathway for longer. We initialize the tiles with damping sampled uniformly in $[0, 10]$. For the 7-DoF robot arm environments, we initialize the target to randomly appear within $(-0.1, -0.2, 0.5), (0.4, 0.2, -0.5)$ in 3D. The position of the target is perturbed with a Gaussian noise sampled from $\mathcal{N}(0, 0.1)$ in each coordinate at each step. We bound the position of the target so that it is confined within the arm's reach. The episode length of all these environments are capped at 1,000.

We build the multi-value networks and meta-baseline on top of the TRPO implementation by OpenAI (Dhariwal et al., 2017). We turned off the GAE enhancement by using $\lambda = 1$ for fair comparison. We found that it makes only a small performance difference (within $\pm 5\%$ using $\lambda = \{0.95, 0.96, 0.97, 0.98, 0.99, 1\}$) in our environments. We use $\gamma = 0.99$ for all three environments. The policy network has two hidden layers, with 128 and 64 hidden neurons on each. The activation function is ReLU (Nair & Hinton, 2010). The KL divergence constraint $\delta$ is 0.01. The learning rate for value functions is $1^{-3}$. The hyperparameter of training the meta-baseline is the same as the discrete-action case.

## L   INPUT-DEPENDENT BASELINES WITH PPO

Figure 8 shows the results of applying input-dependent baselines on PPO (Schulman et al., 2017) in MuJoCo (Todorov et al., 2012) environments. We make three key observations. First, compared to Figure 4 the best performances of PPO in these environments (blue curves) are better than that of TRPO. This is as expected, because the variance of the reward feedbacks in these environments is generally large and the reward clipping in PPO helps. Second, input-dependent baselines improve the policy performance for all environments. In particular, the meta-learning approach achieves the best performance, as it is not restricted to a fixed set of input sequences during training (§5). Third, the trend of learning curve is similar to that in TRPO (Figure 4), which shows our input-

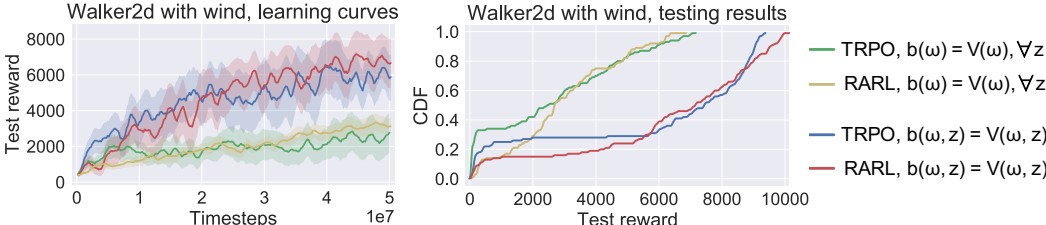

**Figure 9:** The input-dependent baseline technique is complementary and orthogonal to RARL (Pinto et al., 2017). The implementation of input-dependent baseline is MAML (§5). Left: learning curves of testing rewards; shaded area spans one standard deviation; the input-dependent baseline improves the policy optimization for both TRPO and RARL, while RARL improves TRPO in the Walker2d environment with wind disturbance. Right: CDF of testing performance; RARL improves the policy especially in the low reward region; applying the input-dependent baseline boosts the performance for both TRPO and RARL significantly (blue, red).

dependent baseline approach is generally applicable to a range of policy gradient based methods (e.g., A2C (§6.2), TRPO (§6.1), and PPO).

## M    INPUT-DEPENDENT BASELINES WITH RARL

Our work is orthogonal and complementary to adversarial and robust reinforcement learning (e.g., RARL (Pinto et al., 2017)). These methods seek to improve policy robustness by co-training an adversary to generate a worst-case noise process, whereas our work improves policy optimization itself in the presence of inputs like noise. Note that if an adversary generates high-variance noise, similar to the inputs we consider in our experiments (§6), techniques such RARL alone are not adequate to train good controllers.

To empirically demonstrate this effect, we repeat the Walker2d with wind experiment described in §6.1. In this environment, we add a noise (of the same scale as the original random walk) on the wind and co-train an adversary to control the strength and direction of this noise. We follow the training procedure described in RARL (Pinto et al., 2017, §3.3).

Figure 9 depicts the results. With either the standard state-dependent baseline or our input-dependent baseline, RARL generally improves the robustness of the policy, as RARL achieves better testing rewards especially in the low reward region (i.e., compared the yellow curve to green curve, or red curve to blue curve in CDF of Figure 9). Moreover, input-dependent baseline significantly improves the policy optimization, which boosts the performance of both TRPO and RARL (i.e., comparing the blue curve to the green curve, and the red curve to the yellow curve). Therefore, in this environment, the input-dependent baseline helps improve the policy optimization methods and is complementary to adversarial RL methods such as RARL.

## N    INPUT-DEPENDENT BASELINES WITH META-POLICY ADAPTATION

There has been a line of work focusing on fast policy adaptation (Clavera et al., 2018a;b; Harrison et al., 2017). For example, Clavera et al. (2018b) propose a model-based meta-policy optimization approach (MB-MPO). It quickly learns the system dynamics using supervised learning and uses the learned model to perform virtual rollouts for meta-policy adaptation. Conceptually, our work differs because the goal is fundamentally different: our goal is to learn a single policy that performs well in the presence of a stochastic input process, while MB-MPO aims to quickly adapt a policy to new environments.

It is worth noting that the policy adaptation approaches are well-suited to handling model discrepancy between training and testing. However, in our setting, there exists no model discrepancy. In particular, the distribution of the input process is the same during training and testing. For example, in our load balancing environment (Figure 1a, §6.2), the exogenous workload process is sampled from the same distribution during training and testing.

Therefore our work is conceptually complementary to policy adaptation approaches. Since some of these methods require a policy optimization step (e.g., (Clavera et al., 2018b, §4.2)), our input-dependent baseline can help these methods by reducing variance during training. We perform an

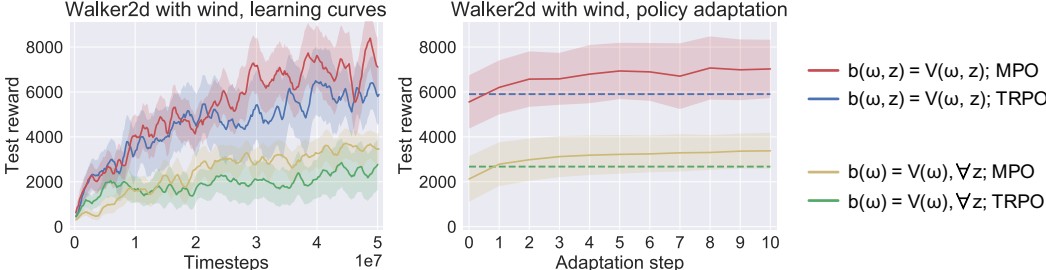

**Figure 10:** The input-dependent baseline technique is complementary to MPO (Clavera et al., 2018b). The implementation of input-dependent baseline is MAML (§5). Left: learning curves in the testing Walker2d environment with wind disturbance; MPO is tested with adapted policy in each testing instance of the wind input; shaded area spans one standard deviation; the input-dependent baseline improves the policy optimization for both TRPO and MPO, while MPO improves TRPO. Right: meta policy adaptation at training timestep $5e7$; adapting the policy in specific input instances help boosting the performance (comparing yellow with green, and red with blue); applying input-dependent baseline generally improves the policy performance.

experiment to investigate this. Specifically, we apply the meta-policy adaptation technique proposed by Clavera et al. (2018b) in our Walker2d environment with wind disturbance (Figure 1c, §6.1). For this environment, although the wind pattern is drawn from the same stochastic process (random walk), we aim to adapt the policy to each particular instantiation of the wind.

Operationally, to reduce complexity, we bypass the supervised learning step for the system dynamics and use the simulator to generate rollouts directly, since the interaction with the simulator is not costly for our purpose and the state transition in our environment is not deterministic. Following the meta-policy adaptation approach, the policy optimization algorithm is TRPO (Schulman et al., 2015a). The meta-policy adaptation algorithm is MAML (Finn et al., 2017). In particular, we performed ten gradient steps to specialize the meta-policy for each instantiation of the input process. For input-dependent baseline, we inherit our meta-baseline approach from §5. Similar to policy adaptation, we adapt our meta-baseline alongside with the policy adaptation in the ten gradient steps for each input instance.

The results of our experiment is shown in Figure 10. The learning curve (left figure) shows the policy performance for 100 unseen test input sequences at each training checkpoint. We measure the performance of MPO after ten steps of policy adaptation for each of the 100 input sequences. As expected, policy adaptation specializes to the particular instance of the input process and improves policy performance in the learning curve (e.g., MPO improves over TRPO, as shown by the green and yellow learning curve). However, policy adaptation does not solve the problem of variance caused by the input process, since the policy optimization step within policy adaptation suffers from large variance. Using an input-dependent baseline improves performance both for TRPO and MPO. Indeed, MPO trained with the input-dependent baseline (and adapted for each input sequence) outperforms the single TRPO policy, as shown by the red learning curve.

This effect is more evident in the policy adaptation curve (right figure). The policy adaptation curve shows the testing performance of the adapted policy at each adaptation step (the meta-policy is taken from the $5e7$ training timestep). With an input-dependent baseline, the meta policy already performs quite well at the $0^{\text{th}}$ step of policy adaptation (without any adaptation). This is perhaps unsurprising, since a single policy (e.g., the TRPO policy trained with input-dependent baseline) can achieve good performance in this environment. However, specializing the meta-policy for each particular input instance further improves performance.

