# OpenReview forum: "Variance Reduction for Reinforcement Learning in Input-Driven Environments"
_ICLR.cc/2019/Conference_

### Official Review · AnonReviewer1 · 2018-11-02
**Interesting research problem (input-driven MDPs), but I think they are missing the most interesting and practically relevant scenario.**

**Rating:** 6
**Confidence:** 4

**Review:**



Summary: This work considers the problem of learning in input-driven environments -- which are characterized by an addition stochastic variable z that can affect the dynamics of the environment and the associated reward the agent might see. The authors show how the PG theorem still applied for a input-aware critic and then they show that the best baseline one can use in conjecture with this critic is a input-dependent one. My main concerns are highlighted in points (3) and (4) in the detailed comments below.

Clarity: Generally it reads good, although I had to go back-and-forth between the main text and appendix several times to understand the experimental side. Even with the supplementary material, examples in Section 3 and Sections 6.2 could be improved in explanation and discussion.

Originality and Significance: Limited in this version, but could be improved significantly by something like point (3)&(4) in detailed comments. Fairly incremental extension of the PG (and TRPO) with the conditioning on the potentially (unobserved) input variables. The fact that a input-aware critic could benefit from a input-aware baseline is not that surprising. The fact that it reduces variance in the PG update is an interesting result; nevertheless I strongly feel the link or comparison needed is with the standard PG update.

Disclaimer: I have not checked the proofs in the appendix.

Detailed comments:

1) On learning the input-dependent baselines: Generalising over context via a parametric functional approximation, like UVFAs [1] seems like a more natural first choice. Also these provide a zero-shot generalisation, bypassing the need for a burn-in period of the task. Can you comment on why something like that was not used at least as baseline?

2) Motivating example. The exposition of this example lacks a bit of clarity and can use some more details as it is not a standard MDP example, so it’s harder to grasp the complexity of this task or how standard methods would do on it and where would they struggle. I think it’s meant to be an example of high variance but the performance in Figure 2 seems to suggest this is actually something manageable for something like A2C. It is also not clear in this example how the comparison was done. For instance, are the value functions used, input-dependent? Is the policy input aware?

3) Input-driven MDP. Case 1/Case 2 : As noted by the authors, in case 1 if both s_t and z_t are observed, this somewhat uninteresting as it recovers a particular structured state variable of a normal MDP. I would argue that the more interesting case here, is where only s_t is observed and z_t is hidden, at least in acting. This might still be information available in hindsight and used in training, but won’t be available ‘online’ -- similar to slack variable, or privileged information at training time.  And in this case it’s not clear to me if this would still result in a variance reduction in the policy update. Case 2 has some of that flavour, but restricts z_t to an iid process. Again, I think the more interesting case is not treated or discussed at all and in my opinion, this might add the best value to this work.

4) Now, as mentioned above the interesting case, at least in my opinion, is when z is hidden. From the formulae(eq. (4),(5)), it seems to be that the policy is unaware of the input variables. Thus we are training a policy that should be able to deal with a distribution of inputs z. How does this compare with the normal PG update, that would consider a critic averaged over z-s and a z-independent baseline? Is the variance of the proposed update always smaller than that of the standard PG update when learning a policy that is unaware of z?

References:
[1] Schaul, T., Horgan, D., Gregor, K. and Silver, D., 2015, June. Universal value function approximators. In International Conference on Machine Learning (pp. 1312-1320).

[POST-rebuttal] I've read the author's response and it clarified some of the concerns. I'm increase the score accordingly.

---

> ### Author Response · Authors · 2018-11-14
> **Author response**
>
> Thank you for the insightful comments!
>
> Regarding these comments:
>
> 1) UVFAs predict values based on specific goals. These methods require taking “goal embedding” explicitly as input. In our formulation of input driven environment, however, there aren’t really different goals in each task. Nonetheless, one can still use similar idea to take the exogenous sequence as an explicit input in the value function, using recurrent neural network structures such as LSTM. We actually did this and reported our findings in the paper, in the beginning of Section 5: “A natural approach to train such baselines is to use models that operate on sequences (e.g., LSTMs). However, learning a sequential mapping in a high-dimensional space can be expensive. We considered an LSTM approach but ruled it out when initial experiments showed that it requires orders of magnitude more data to train than conventional baselines for our environments.” We intend to add an experiment showing the learning curve with an LSTM approach to the appendix.
>
> 2) The point of this example is to show that the variance from the input process can negatively affect the policy, even for an extremely simple task. In this 2-server load balance task, the agent should just learn the simple optimal policy of joining the shortest queue (visualized in Figure 2(c) left). However, the variance in the input sequence makes the PG unable to converge to the optimal. Here, we compared the vanilla A2C with the standard state-only baseline to that with the input-dependent baseline. It is clear that vanilla A2C performs suboptimally (Figure 2(b) right); and this is due to the significant difference in the PG variance in different baselines (Figure 2(b) left, notice the log scale).
>
> The reason that vanilla A2C is ineffective in this example is that the return (total reward) for an action depends on the job arrival sequence that follows that action. For example, if the arrivals consist of a burst of large jobs, the reward (negative number of jobs in the system) will be poor, regardless of the action. We will expand the discussion in Section 3 to provide more details and intuition.
>
> About the input to the baseline and policy: the input-dependent baseline takes state s_t and the entire future input process z_{t:\infty} as input; the state-only baseline only takes s_t as input; in both cases, the policy network takes s_t and z_t (only at time t) as input.
>
> 3 and 4) Thank you for this interesting comment. We focused on the two cases in Figure 3 mainly because they result in fully observable MDPs, and in many applications of interest, the input is readily observable. However, the scenario in which the input z_t is not observed is indeed also interesting. This case results in a POMDP.
>
> Input-dependent baselines reduce variance in the POMDP case as well. Our results (e.g., Theorems 1 and 2)  also apply to this setting. In fact, in the POMDP case, the input process does not even need to be Markov; it can be any general stochastic process that does not depend on the states and actions.
>
> Intuitively, the reason is that much of the variance in PG in input-driven environments is caused by the variance in the input sequence that follows an action. For example, in the windy walker environment (Figure 1c), it is the entire sequence of wind after step t that affects the total reward, not just the wind observation at time t. As a result, regardless of whether or not the input is observed at each step t, using the entire input sequence in the baseline reduces variance.
>
> Interestingly, the HalfCheetah with floating tiles environment (Figure 1d) is actually a POMDP---the agent only observes the torques of the cheetah’s body but not the buoyancy of the tiles. As shown in Figure 4 (middle), our technique helped reduce variance and improve PG performance. Also, we re-ran our experiments on the Walker2d with wind environment without providing z (the wind) to the policy. The results show that our input-dependent baseline improves the policy performance similar to the case where z is observed. We will shortly add this result to the paper.
>
> In summary, we are making the following changes to the paper. We will add a case for POMDP to Figure 3, and discuss the derivation for the POMDP (which is almost identical to the MDP case). We will also include the POMDP version of Walker2d with wind result.
>
> We also realize that the notation was confusing. As mentioned in the 2nd paragraph of page 5, we were using s_t to denote the tuple (s_t, z_t) in the derivations. We will improve the notation by explicitly defining the observed signal, o_t, used by the policy in each case. For the MDP case, o_t = (s_t, z_t). For the POMDP case, o_t = s_t.

---

### Official Review · AnonReviewer2 · 2018-11-03
**Strong paper for environment in which outcomes are strongly influenced by exogenous factors**

**Rating:** 9
**Confidence:** 4

**Review:**

The paper introduces and develops the notion of input-dependent baselines for Policy Gradient Methods in RL.

The insight developed in the paper is clear: in environments such as data centers or outside settings external factors (traffic load or wind) constitute high magnitude perturbations that ultimately strongly change rewards.
Learning an input-dependent baseline function helps clear out the variance created by such perturbations in a way that does not bias the policy gradient estimate (the authors provide a theoretical proof of that fact).

The authors propose different methods to train the input dependent baseline function:
   o) a multi-value network based approach
   o) a meta-learning approach
The performance of these two methods is compared on simulated robotic locomotion tasks as well as a load balancing and video bitrate adaptation task.
The input dependent baseline strongly outperforms the state dependent baseline in both cases.

Strengths:
   o) The paper is well written
   o) The method is novel and simple while strongly reducing variance in Monte Carlo policy gradient estimates without inducing bias.
   o) The experiment evidence is strong.

Weaknesses:
   o) Vehicular traffic has been the subject of recent development through deep reinforcement learning (e.g. https://arxiv.org/pdf/1701.08832.pdf and https://arxiv.org/pdf/1710.05465.pdf). In this particular setting exogenous noise (demand for throughput and accidents) could strongly benefit from input dependent baselines. I believe the authors should mention such potential applications of the method which may have major societal impact.
   o) There is a lot of space dedicated to well know facts about policy gradient methods. I believe it could be more impactful to put the proof of Theorem 1 in the main body of the paper as it is clearly a key theoretical property.

---

> ### Author Response · Authors · 2018-11-14
> **Author response**
>
> We appreciate your encouraging comments!
>
> We agree that the traffic control environment is a perfect fit for the techniques we proposed. Thanks for the suggestions and the pointers to the existing simulators---we will mention these potential applications in the introduction/conclusions.
>
> In our submission, we moved the proofs to appendix due to space constraints. We will trim down the text of the facts about PG methods to clear up rooms for the proof of Theorem 1.

---

### Official Review · AnonReviewer4 · 2018-11-09
**Interesting premise, needs more clarity/comparisons**

**Rating:** 7
**Confidence:** 4

**Review:**


Introduction:
“Since the state dynamics and rewards depend on the input process” -> why do the rewards depend on the input process conditioned on the state?

Does the scenario being considered basically involve any scenario with stochastic dynamics? Or is the fact that the disturbances may come from a stateful process what makes this distinct?

if the input sequence following the action -> vague, would help if this would just be written a bit more clearly.

Is just the baseline input dependent or does the policy need to be input dependent as well? From later reading, this point is still quite confusing. One line says “At time t, the policy only depends only on (st, zt).”. Another line says that the policy is pi_theta(a|s), with no mention of z. I’m pretty confused by the consistency here. This is also important in the proof of Lemma 1, because P(a|s,z) = pi_theta(a|s). Please clarify this.

Section 4:
 Is the IID version of Figure 3 basically the same as stochastic dynamics? (Case 2)

Section 4.1
“In input-driven MDPs, the standard input-agnostic baseline is ineffective at reducing variance” -> can you give some more intuition/proof as to why.

In Lemma 2, how come the Q function is dependent on z, but the policy is only dependent on s (not even the current and past z’s).

I think the proof of theorem 1 should be included in the main paper rather than unnecessary details about policy gradient.

Theorem 1 and theorem 2 are really some of the most important parts of the paper, and they deserve a more thorough discussion besides the 2 lines that are in there right now.


Algorithm 1 -> should it be eqn 4?

The meta-algorithm provided in Section 5 is well motivated and well described. An experimental result including what happens with LSTM baselines would be very helpful.

One question is whether it is actually possible to know what the z’s are at different steps? In some cases these might be latent and hard to infer?

Can you compare to Clavera et al 2018? It seems like it might be a relevant comparison.

The difference between MAML and the 10 value network seems quite marginal. Can the authors discuss why this is? And when we would expect to see a bigger difference.

Related work: Another relevant piece of work
Meta-Learning Priors for Efficient Online Bayesian Regression

Major todos:
1. Improve clarity of what z's are observed, which are not and whether the policy is dependent on these or not.
2. Compare with other prior work such as Clavera et al, Harrison et al.
3. Add more naive baselines such as training an LSTM, etc.
4. Provide more analysis of the meta-learning component, how much does it actually help.

Overall impression:  I think this paper covers an interesting problem, and proposes a simple, straightforward approach conditioning the baseline and the critic on the input process. What bothers me in the current version of the paper is the lack of clarity about the observability of z, where it comes from and also some lack of comparisons with other prior methods. I think these would make the paper stronger.

---

> ### Author Response · Authors · 2018-11-14
> **Author response**
>
> Thank you for the constructive comments!
>
> We first address the major comments and then respond to the detailed questions in a separated comment.
>
> 1. [What is observed?] During policy inference at each MDP step t, the agent observes s_t and z_t (the current value of the input process). Therefore the policy can depend on the current observed value of the input z_t, but not on the future input sequence z_{t:\infty} (which has not yet happened). At training time, however, the baseline computation for step t depends on the entire future sequence z_{t:\infty}. As explained in the beginning of Section 4.1, this is possible because the entire input sequence is known at training time.
>
> We realize that the notation was confusing. As mentioned in the 2nd paragraph of page 5, we use s_t to denote the tuple (s_t,z_t) for the derivations. We will improve the notation by explicitly defining the observed signal, o_t = (s_t, z_t), which the policy takes as input at each step t.
>
> 2. [Additional comparisons to prior work] Policy adaptation approaches like Clavera et al. learn a “meta-policy” that can be quickly adapted for different environments. By contrast, our goal is to learn a single policy that performs well in the presence of a stochastic input process. In other words, we are improving policy optimization itself in environments with stochastic inputs. We do not consider transfer of a policy trained for one environment to another. In terms of training a common policy, our work is more related to RARL (Pinto et al.), which we discuss and compare with in Appendix L.
>
> It is worth noting that approaches like Clavera et. al. are well-suited to handling model discrepancy between training and testing. However, in our setting, there isn’t any model discrepancy. In particular, the distribution of the input process is the same during training and testing. Nonetheless, our work shows that standard policy gradient methods have difficulty in input-driven environments, and input-dependent baselines can substantially improve performance.
>
> Therefore our work is orthogonal and complementary to policy adaptation approaches. Since some of these methods require a policy optimization step (e.g., Section 4.2 of Clavera et al. 2018), our input-dependent baseline can help these methods by reducing variance during training. Appendix L shows an example of such improvements for RARL. We will try to also add an example for a policy adaptation approach.
>
> 3. [The LSTM method for learning input-dependent baselines] LSTM suffers from unnecessarily high complexity in training. In our experiments, we considered an LSTM approach but ruled it out when initial experiments showed that it requires orders of magnitude more data to train than conventional baselines for our environments (cf. beginning of Section 5). We will add the learning curves with LSTM baseline in the appendix.
>
> 4. [The meta-learning baseline] The actual performance gain for a meta-learned baseline over a multi-value-network is environment-specific. Conceptually, the multi-value-network falls short when the task requires training with a large number of input instantiations to generalize to new input instances. We have not analyzed how policy quality varies with the number of input instantiations considered during training. However, we expect that this depends on a variety of factors, such as the distribution of the input process (e.g., from a large deviations standpoint); the time horizon of the problem; the relative magnitude of the variance due to the input process compared to other sources of randomness (e.g., actions). The advantage of the meta-learning approach compared to the multi-value network approach is that we can train with an unbounded number of input instantiations. We will add this discussion to Section 5.

---

> > ### Author Response · Authors · 2018-11-14
> > **Responses to other questions**
> >
> >
> > -- Why do the rewards depend on the input process conditioned on the state?
> >
> > To clarify, by “state dynamics and rewards depend on the input process,” we mean that the input process can affect the rewards because it affects the state transitions. However, our model indeed covers the general case, in which the reward might depend on both the state and the input. For example, consider a robotics task in which the reward is the speed of the robot, the state is the current position of the robot’s joints,   and the input is an external force applied to the robot at each step. The speed of the robot (reward) depends on the force (input) even with knowledge of its current position (state).
> >
> > -- What makes the input process we considered distinct from any stochastic dynamics?
> >
> > The main distinction here is that the input process must be “exogenous,” i.e. it doesn’t depend on the state and actions; see the graphical models in Figure 3. This property is necessary for the input-dependent baseline to not introduce bias.
> >
> > -- A strong action could end up with a lower-than-average return if the input sequence following the action is unfavorable -> vague
> >
> > This sentence was trying to give an intuition for why the variance in reward caused by the input process can confuse a policy gradient algorithm. We will rephrase the sentence and explain this better. We will also provide more intuition about this point in Section 3.
> >
> > Consider the load balancing example in Section 3. The return (total reward) for an action depends on the job arrival sequence that follows that action. For example, if the arrivals consist of a burst of large jobs, the reward (negative number of jobs in the system) will be poor, regardless of the action. We will add this intuition to Section 3.
> >
> > -- Is just the baseline input dependent or does the policy need to be input dependent as well?
> >
> > The baseline depends on the sequence of input values z_{t:\infty}, but the policy can only depend on the input observed at the current step t.  Note that the policy cannot depend on the future input values, since at time t, the agent has no way of knowing z_{t+1,\infty}.
> >
> > -- “In input-driven MDPs, the standard input-agnostic baseline is ineffective at reducing variance” -> can you give some more intuition/proof as to why.
> >
> > As mentioned above, we will add more intuition for this to Section 3.
> >
> > -- More discussions about theorem 1 and 2.
> >
> > Thanks for the suggestion! We will trim the discussion of policy gradient and include the proof of theorem 1.
> >
> > -- Algorithm 1 should use eqn 4.
> >
> > Yes, it is more appropriate to refer to Equation 4 in Algorithm 1. We will use this.
> >
> > -- Is it possible to know z at each step? What if z is not observable and hard to infer
> >
> > In many applications, the input process is naturally observable to the agent. For example, in most computer systems applications, the inputs to the environments (e.g., network bandwidth, workload) are measured or readily observed. However, even if the agent does not observe the input at each step, our proposed approach (multi-value-network and meta-learning) can still work as long as we can repeat the same input sequence during training. As discussed in Section 5, this can be done with a simulator (e.g., control the wind in MuJoCo simulator) or by repeating input sequences (e.g., repeat the same workload for a load balancing agent) in an actual system. For future work, we think that investigating efficient architectures for input-dependent baselines for cases where the input process cannot be controlled in training is an interesting direction.
> >
> > -- Meta Learning Priors for Efficient Online Bayesian Regression
> >
> > Thank you for the suggestion. This is a relevant piece of work on applying meta learning for faster adaptation of GP regression. We will add it in the related work session.

---

### Author Response · Authors · 2018-11-22
**Paper update**

We again thank all reviewers for their comments, and have updated our paper accordingly.

Specifically, the major changes are:
   - In §4.1, we improved the clarity of our notations by explicitly defining the observation \omega_t at each time t. We used \omega_t instead o_t because the letter o is visually too similar to a. We updated our theorems and proofs using this notation.
   - We extended the case 2 of input-driven MDP to include the POMDP case (Figure 3b), and have showed all our derivation and conclusions apply.
   - We added a comparison to the meta-policy optimization approach (Clavera et al. 2018) in Appendix N.
   - In addition to mentioning our findings with LSTM in §5, we also added the corresponding learning curves in appendix G.
   - We updated our motivating example (§3) to give a better intuition.
   - We shortened the policy gradients description in the introduction and background sections, and moved the proof of theorem 1 into the main text in §4.1.
   - In §5, we added a discussion of when we expect the gain of MAML to further exceed that of the multi-value-network approach.

Please let us know if you have further comments. Thanks!

---

### Meta-Review · Area_Chair1 · 2018-12-13

**Confidence:** 4
**Recommendation:** Accept (Poster)

**Metareview:**

This paper proposes an input-dependent baseline function to reduce variance in policy gradient estimation without adding bias. The approach is novel and theoretically validated, and the experimental results are convincing. The authors addressed nearly all of the reviewer's concerns. I recommend acceptance.